# EGCG Promotes FUS Condensate Formation in a Methylation-Dependent Manner

**DOI:** 10.3390/cells11040592

**Published:** 2022-02-09

**Authors:** Aneta J. Lenard, Qishun Zhou, Corina Madreiter-Sokolowski, Benjamin Bourgeois, Hermann Habacher, Yukti Khanna, Tobias Madl

**Affiliations:** 1Gottfried Schatz Research Center for Cell Signaling, Metabolism and Ageing, Molecular Biology and Biochemistry, Medical University of Graz, 8010 Graz, Austria; aneta.lenard@medunigraz.at (A.J.L.); qishun.zhou@medunigraz.at (Q.Z.); corina.madreiter@medunigraz.at (C.M.-S.); benjamin.bourgeois@medunigraz.at (B.B.); hermann.habacher@medunigraz.at (H.H.); yukti.khanna@medunigraz.at (Y.K.); 2BioTechMed-Graz, 8010 Graz, Austria

**Keywords:** EGCG, FUS, RG/RGG, LLPS, neurodegenerative diseases, arginine methylation

## Abstract

Millions of people worldwide are affected by neurodegenerative diseases (NDs), and to date, no effective treatment has been reported. The hallmark of these diseases is the formation of pathological aggregates and fibrils in neural cells. Many studies have reported that catechins, polyphenolic compounds found in a variety of plants, can directly interact with amyloidogenic proteins, prevent the formation of toxic aggregates, and in turn play neuroprotective roles. Besides harboring amyloidogenic domains, several proteins involved in NDs possess arginine-glycine/arginine-glycine-glycine (RG/RGG) regions that contribute to the formation of protein condensates. Here, we aimed to assess whether epigallocatechin gallate (EGCG) can play a role in neuroprotection via direct interaction with such RG/RGG regions. We show that EGCG directly binds to the RG/RGG region of fused in sarcoma (FUS) and that arginine methylation enhances this interaction. Unexpectedly, we found that low micromolar amounts of EGCG were sufficient to restore RNA-dependent condensate formation of methylated FUS, whereas, in the absence of EGCG, no phase separation could be observed. Our data provide new mechanistic roles of EGCG in the regulation of phase separation of RG/RGG-containing proteins, which will promote understanding of the intricate function of EGCG in cells.

## 1. Introduction

Neurodegenerative diseases (NDs) affect millions of people worldwide and therefore constitute a major public health threat. NDs occur when nerve cells in the brain or peripheral nervous system lose their function over time, leading to cellular death [1]. The hallmarks of these diseases, which encompass Alzheimer’s disease (AD), Parkinson’s disease (PD), amyotrophic lateral sclerosis (ALS), and frontotemporal dementia (FTD), are the formation of pathological aggregates and the presence of abnormal protein deposits in neural cells [2,3,4,5]. Until today, and to our knowledge, no effective treatment has been reported to cure or slow down the progression of these diseases [6,7].

Several studies have demonstrated the role of green tea catechins in neuroprotection, with epigallocatechin gallate (EGCG) being widely studied because it constitutes 65% of all catechins in green tea [8,9,10,11]. The green tea catechins play a variety of roles in neuroprotection, such as exerting (i) anti-oxidative properties through radical scavenging and metal ion chelation [12,13,14], (ii) anti-apoptotic properties through the reduction in pro-apoptotic gene expression [15], (iii) anti-inflammatory properties through the inhibition of microglia activation [16], and (iv) anti-amyloidogenic properties through the remodeling of toxic aggregates [8,17]. The latter role of catechins in neuroprotection is of great interest as it counteracts the hallmark of NDs (formation of protein aggregates), which is thought to drive the process of neurodegeneration. Indeed, EGCG has been shown to directly interact with various amyloidogenic proteins, such as amyloid-β or α-synuclein, involving their low-complexity domains, thus inhibiting their propensity to form toxic aggregates [18,19,20,21,22]. Several amyloidogenic proteins harbor, in addition to their amyloidogenic domain (AD), regions rich in arginine and glycine residues (RG/RGG), and these regions contribute to their propensity to phase separate and possibly aggregate in (patho)physiological conditions [23,24,25,26,27,28]. Given that EGCG possesses several aromatic rings that could mediate cation–π interactions with arginines of RG/RGG regions, we hypothesized that EGCG directly interacts with the RG/RGG region of these amyloidogenic proteins, modulates the formation of biomolecular condensates, and in turn contributes to the neuroprotective effect of EGCG.

To this end, we chose fused in sarcoma (FUS) as a model protein. FUS contains a N-terminal SYGQ-rich domain, with prion-like properties, as well as several RG/RGG-rich regions in its C-terminal disordered tail (Figure 1A). Both the prion-like domain and the RGG3-PY region of FUS have been shown to contribute to FUS phase separation and stress granule (SG) association. Aggregation and/or fibril formation of FUS has been linked to several (patho)physiological conditions [23,27,29]. For example, FUS cytoplasmic aggregates were found in neuronal cells of ALS and FTD patients and proposed to drive neurodegeneration. Therefore, we asked the question of whether EGCG can directly bind to the RGG3-PY region of FUS (FUS^RGG3PY^), thereby modulating its propensity to phase separate.

Here, we used nuclear magnetic resonance (NMR) spectroscopy to obtain molecular details on EGCG binding to FUS and its regulation by FUS arginine methylation and RNA binding. We used turbidity assays and differential contrast microscopy (DIC) in order to decipher the role of EGCG, RNA, and arginine methylation on FUS condensate formation. We found that EGCG directly interacts with the RGG3-PY region of FUS in a non-specific and arginine methylation-independent manner, leading to the formation of FUS condensates at high micromolar EGCG concentrations. We uncovered that RNA can directly interact with both methylated and unmethylated FUS^RGG3PY^ and that FUS phase separation was inhibited by methylation. Unexpectedly, arginine methylation of FUS enhanced its EGCG affinity by several orders of magnitude, and EGCG restored RNA-driven phase separation of methylated FUS at low micromolar concentrations.

In summary, our data provide unexpected evidence that EGCG can compensate for the inhibitory effect of arginine methylation on RNA-driven phase separation of RG/RGG regions. Although further studies are required to investigate the physiological significance of EGCG in the regulation of condensate formation, our studies shed light on the new mechanistic functions of EGCG that will promote a better understanding of the various roles of EGCG in human cells.

## 2. Materials and Methods

### 2.1. Recombinant Protein Expression and Purification

Recombinant His_6_-protein A-tagged FUS^RGG3PY^ (amino acids 454–526), His_6_-protein A-tagged FOXO4^CR3^ (amino acids 454–505), and His_6_-protein A-tagged axin-1 (amino acids 351–500) containing a tobacco etch virus (TEV) protease cleavage site after protein A were expressed from codon-optimized synthetic genes inserted into pETM11-based vectors (Genscript, Piscataway, NJ, USA). For the expression of recombinant protein, the construct was transformed into *E. coli* BL21(DE3) Star cells, and grown in a standard lysogeny broth (LB) medium (Carl Roth GmbH, Karlsruhe, Germany) at 37 °C until an optical density (OD 600 nm) value has reached 0.6–0.8. Then, cells were induced with 1 mM isopropyl β-d-1-thiogalactopyranoside (IPTG), (BLD Pharmatech GmbH, Kaiserslautern, Germany) and grown for 16 h at 20 °C. For the purpose of NMR experiments, 10 mL overnight precultures were transferred to minimal media (100 mM KH_2_PO_4_, 50 mM K_2_HPO_4_, 60 mM Na_2_HPO_4_, 14 mM K_2_SO_4_, 5 mM MgCl_2_, pH 7.2 adjusted with HCl and NaOH with 0.1 dilution of trace element solution (41 mM CaCl_2_, 22 mM FeSO_4_, 6 mM MnCl_2_, 3 mM CoCl_2_, 1 mM ZnSO_4_, 0.1 mM CuCl_2_, 0.2 mM (NH_4_)_6_Mo_7_O_24_, 17 mM EDTA) (all chemical compounds here and below, if not further specified, were purchased from Carl Roth GmbH, Karlsruhe, Germany. FeSO_4_, ZnSO_4_ CuCl_2_ and (NH_4_)_6_Mo_7_O_24_ from VWR International bvba, Leuven Belgium, CoCl_2_ from AppliChem GmbH - An ITW Company, Darmstadt, Germany) supplemented with 1 g of ^15^NH_4_Cl (Eurisotop, Saint-Aubin, France), and either with 6 g of unlabeled glucose or 2 g of ^13^C_6_H_12_O_6_ (Cambridge Isotope Laboratories, Tewksbury, MA, USA), followed by a growth as described for unlabeled protein. Cells were harvested (6,000 rpm for 10 min at 4 °C), transferred to a denaturing lysis buffer (50 mM Tris-HCl pH 7.5, 150 mM NaCl, 20 mM imidazole, 6 M urea), and sonicated (70% amplitude, 1 s pulse for 12 min on ice bath with Qsonica MC-18 sonicator (Qsonica, Newtown, CT, USA). His_6_-protein A-tagged FUS^RGG3PY^, FOXO4^CR3^ and axin-1 were purified using nickel-nitrilotriacetic (Ni-NTA) agarose resin (Carl Roth GmbH, Karlsruhe, Germany) and eluted into buffer containing 50 mM Tris-HCl pH 7.5, 1 M NaCl, 500 mM imidazole, 2 mM tris(2-carboxyethyl)phosphine (TCEP) (BLD Pharmatech GmbH, Mehlingen, Germany), 0.04% NaN_3_. The eluted protein was desalted to buffer containing 50 mM Tris-HCl pH 7.5, 150 mM NaCl, 20 mM imidazole, 2 mM TCEP, 0.04% NaN_3_, and subjected to overnight TEV treatment at 4 °C. The cleaved FUS^RGG3PY^, FOXO4^CR3,^ or axin-1 were then isolated by a second affinity purification using Ni-NTA beads. A final size-exclusion chromatography purification step was performed in the buffer of interest with the use of Superdex 75 Increase 10/300 GL column (GE Healthcare Bio-Sciences AB, Uppsala, Sweden) at room temperature. Fractions corresponding to FUS^RGG3PY^, FOXO4^CR3^, or axin-1 were identified by SDS PAGE gel and either used immediately for experiments or stored at −80 °C until further use.

Recombinant rat His_6_-PRMT1 (amino acids 11–353) was inserted into a pET28b-His_6_ vector (Novagen, Merck, Darmstadt, Germany), and the expression has been previously described [30]. The expression construct was transformed into *E. coli* BL21(DE3) star cells, and 10 mL of overnight preculture was transferred to 1 L of LB medium. The expression culture was grown at 37 °C and at 160 rpm, and cells were induced at an OD 600 nm of 0.6–0.8 with 1 mM IPTG, followed by protein expression for 16 h at 20 °C and at 160 rpm. Cells were harvested (6,000 rpm for 10 min at 4 °C) and lysed by sonication (70% amplitude, 1 s pulse for 12 min on ice bath with Qsonica MC-18 sonicator in a non-denaturing lysis buffer (50 mM Tris-HCl pH 7.5, 150 mM NaCl, 20 mM imidazole, 2 mM TCEP, 10% (*v*/*v*) glycerol). Uncleaved His_6_-PRMT1 was then purified using 5 mL HisTrap HP column (Cytiva Europe GmbH, Vienna, Austria) at 4 °C and eluted with 10 column volumes into buffer containing 50 mM Tris-HCl pH 7.5, 1 M NaCl, 500 mM imidazole, 2 mM TCEP, 0.04% NaN_3_. The eluted His_6_-PRMT1 was desalted into methylation buffer (50 mM Na_2_HPO_4_/NaH_2_PO_4_ pH 8.0, 150 mM NaCl, 2 mM dithiothreitol (DTT, AppliChem GmbH, Darmstadt, Germany), 0.04% NaN_3_) using a HiPrep 26/10 Sephadex G-25 desalting column (GE Healthcare Bio-Sciences AB, Uppsala, Sweden) and stored overnight at 4 °C. As a final polishing step, size-exclusion chromatography purification was performed in the methylation buffer using a Superdex 200 Increase 10/300 GL column (GE Healthcare Bio-Sciences AB, Uppsala, Sweden) at 4 °C. Fractions corresponding to PRMT1 were identified by SDS PAGE gel and used immediately for experiments.

The in vitro methylation reaction of FUS^RGG3PY^ was performed as follows. For the reaction, respective size-exclusion fractions of FUS^RGG3PY^ and PRMT1 were eluted into the methylation buffer and collected. FUS^RGG3PY^ was in vitro methylated by incubation with PRMT1 in the presence of 2 mM S-adenosyl-L-methionine (SAM) at room temperature for 24 h, and PRMT1 was used at a molar ratio of 1:2 for FUS^RGG3PY^. In vitro methylated FUS^RGG3PY^ (meFUS^RGG3PY^) was then isolated from PRMT1 by heating for 10 min at 95 °C and applying the sample on a size-exclusion chromatography column in the buffer of interest (Superdex 75 Increase 10/300 GL).

### 2.2. Turbidity Assay

For turbidity assays, FUS^RGG3PY^ and meFUS^RGG3PY^ samples were prepared in 50 mM sodium succinate pH 6.0, 150 mM NaCl, 2 mM TCEP, 0.04% NaN_3_. Turbidity measurements were conducted at 620 nm in 96-well plates with 90-μL samples with the use of a CLARIOstar Plate Reader (BMG LABTECH GmbH, Ortenberg, Germany).

### 2.3. Differential Interference Contrast Microscopy

FUS^RGG3PY^ and meFUS^RGG3PY^ samples were prepared in 50 mM sodium succinate pH 6.0, 150 mM NaCl, 2 mM TCEP, 0.04% NaN_3_. The 30-μL sample was plated on a 30 mm No. 1 round glass coverslip and mounted on an Observer D1 microscope with 100×/1.45 oil immersion objective (Zeiss, White Plains, NY, USA). Protein droplets were viewed using a HAL100 halogen lamp, and images were captured with an OrcaD2 camera (Hamamatsu, Japan) using VisiView 4.0.0.13 software (Visitron Systems GmbH, Puchheim, Germany). The formation of droplets was induced by the addition of RNA/EGCG to all protein samples, and pictures were recorded for a duration of 1 h after the addition of RNA/EGCG.

### 2.4. NMR Spectroscopy

All NMR experiments were conducted at 25 °C on a Bruker 600 MHz spectrometer (Bruker Biospin, Rheinstetten, Germany) equipped with a TXI probe equipped with z-axis gradients and using between 30 and 600 µM of [U-^15^N]- or [U-^15^N,^13^C]-labeled FUS^RGG3PY^. All spectra were processed using TopSpin 4.0.9, and heteronuclear spectra were analyzed with the use of NMRFAM-Sparky 3.114, [31] and CcpNMR 2.4.2 [32] software. All experiments, except axin-1 and FOXO4^CR3^ titrations, were performed using protein samples prepared in 50 mM sodium succinate pH 6.0, 150 mM NaCl, 2 mM TCEP, 0.04% NaN_3_, and 10% (*v*/*v*) deuterium oxide (Eurisotop, Saint-Aubin, France) was added for the lock signal in all samples. Axin1-EGCG titrations were performed in 50 mM NaH_2_PO_4_/Na_2_HPO_4_ pH 6.5, 150 mM NaCl, 2 mM DTT and 10% (*v*/*v*) deuterium oxide. FOXO4 CR3-EGCG titrations were performed in 50 mM NaH_2_PO_4_/Na_2_HPO_4_ pH 6.5, 2 mM DTT and 10% (*v*/*v*) deuterium oxide. The following experiments were recorded: ^1^H-^15^N HSQC, ^1^H-^13^C HSQC, ^1^H-^1^H NOESY, heteronuclear ^15^N{^1^H} NOE, (H)CC(CO)NH, CBCA(CO)NH, HN(CA)NNH(N), and HN(CA)NNH(H). The assigned spectra were uploaded to Biological Magnetic Resonance Bank and are available under the accession codes 51314 and 51315. For the chemical shift perturbation (CSP) analysis, normalized values of FUS^RGG3PY^/meFUS^RGG3PY 1^H-^15^N cross-peaks upon binding to RNA and/or EGCG were calculated according to the following formula:(1)$$CSP=\sqrt{{\left(\delta_{H}\right)}^{2}+\frac{{\left(\delta_{N}\right)}^{2}}{10}}$$
where *δ_H_* corresponds to the ^1^H chemical shift difference between FUS^RGG3PY^/meFUS^RGG3PY 1^H-^15^N cross-peaks in the bound and free states, and *δ_N_* denotes the ^15^N chemical shift difference between FUS^RGG3PY^/meFUS^RGG3PY 1^H-^15^N cross-peaks in the bound and free states, respectively.

## 3. Results

We successfully expressed and purified recombinant FUS^RGG3PY^ and PRMT1 to perform in vitro methylation. Using non-methylated FUS^RGG3PY^ and methylated FUS^RGG3PY^ (meFUS^RGG3PY^) samples, we carried out solution-based characterization of the (me)FUS^RGG3PY^ interactions with EGCG and RNA as well as their implications in liquid-liquid phase separation.

### 3.1. EGCG Directly Interacts with the RGG3-PY Region of FUS with High Micromolar Affinity

We firstly aimed to assess whether EGCG can directly interact with the RGG3-PY region of FUS. To obtain residue-resolved information of this putative interaction, we performed NMR spectroscopy-derived titrations of recombinant ^15^N isotope-labeled FUS^RGG3PY^ in the absence or presence of EGCG. Stepwise addition of EGCG resulted in a progressive line broadening and chemical shift perturbation (CSP) of several ^1^H-^15^N FUS^RGG3PY^ cross-peaks (Figure 1B,C). This shows that EGCG can directly interact with the RGG3-PY region of FUS with a binding affinity in the high micromolar to the millimolar range, as suggested by the linear increase in CSPs upon EGCG addition (Figure 1D). A closer inspection of ^1^H-^15^N FUS^RGG3PY^ cross-peaks affected by the addition of EGCG revealed that the entire FUS RGG3-PY region is affected, suggesting several EGCG binding sites (Figure 1B–D).

### 3.2. Arginine Methylation of the RGG3-PY Region of FUS Enhanced Binding to EGCG

Several studies have shown that FUS can be specifically methylated on several arginines by the protein arginine *N*-methyltransferase 1 (PRMT1) [23,33,34]. Here, our objective was to assign methylated arginine residues in FUS^RGG3PY^ using NMR spectroscopy and then assess the effect of FUS^RGG3PY^ arginine methylation on EGCG binding. We, therefore, carried out in vitro arginine methylation of FUS^RGG3PY^ using purified recombinant PRMT1 and tested the effect of arginine methylation on EGCG binding employing NMR spectroscopy.

Successful methylation of several arginines, including several previously published residues [35], was confirmed by NMR spectroscopy using (H)CC(CO)NH experiments (Figure 2A,B). Methylated arginines exhibit a carbon delta (Cδ) resonance shifted by about 2 ppm compared to unmethylated arginines (Figure 2B). This allowed us to cluster arginines in FUS^RGG3^ depending on their degree of methylation in three groups: (a) unmethylated arginines (R472, R498, R503, and R514), (b) partly methylated arginines (R473, R476) (c) methylated arginines (R481, R485, R487, R491, R495) (Figure 2B). Arginines in the FUS^PY^ region remain unmethylated. To investigate the impact of arginine methylation on the local rigidity of FUS^RGG3PY^, we performed ^15^N{^1^H} heteronuclear NOE experiments using isotopically labeled un- and methylated FUS^RGG3PY^ (Appendix A
Figure A1). The observed low heteronuclear NOE values indicate that both un- and methylated FUS^RGG3PY^ are intrinsically disordered, in which methylation status has only negligible effects on the variation of local rigidity. Overall, we identified multiple methylated arginines well-distributed along the RGG3-PY region of FUS.

To decipher whether arginine methylation in FUS^RGG3PY^ alters its binding with EGCG, we performed an NMR-based titration by stepwise addition of EGCG in a solution containing meFUS^RGG3PY^. This resulted in a progressive line broadening (Figure 2C) as well as higher CSPs of several ^1^H-^15^N meFUS^RGG3PY^ cross-peaks, in comparison to FUS^RGG3PY^ (Figure 2D). Interestingly, increased CSPs were mostly found in the region of amino acids 480–500, which corresponds to the region containing methylated arginines. This suggested a higher affinity of EGCG to methylated FUS, resulting from arginine methylation in the RG/RGG region. Overall, our data revealed that PRMT1-mediated arginine methylation of the FUS RGG3-PY region enhances its EGCG binding.

### 3.3. Arginine Methylation Does Not Abrogate RNA or EGCG Binding of FUS^RGG3-PY^

We and others have previously demonstrated that RNA can directly interact with RG/RGG regions of several RNA-binding proteins (RBPs), including FUS, CIRBP, and hnRNP family proteins [23,24,25,36,37]. This prompted us to determine whether (i) arginine methylation of the FUS^RGG3PY^ is compatible with RNA binding and whether (ii) RNA binding affects EGCG binding of methylated or unmethylated FUS^RGG3PY^, respectively. We first tested the effect of FUS^RGG3PY^ arginine methylation on RNA binding using NMR spectroscopy. Stepwise addition of (UG)_12_ RNA to solutions of either ^15^N-labeled methylated or unmethylated FUS^RGG3PY^ caused CSPs of several ^1^H-^15^N (me)FUS^RGG3PY^ cross-peaks (Figure 3A–C). In addition to CSPs, line broadening was observed for unmethylated FUS. The CSPs of meFUS^RGG3-PY^ binding to RNA at 4:1 molar ratio are higher than the CSPs of FUS^RGG3PY^ (Figure 3C). These results indicated that RNA is binding to both unmethylated and methylated RGG3-PY regions of FUS but is stronger with the methylation of arginines. The line broadening observed for unmethylated FUS is likely a result of its propensity to phase separate in the presence of RNA, as previously published [23].

We then performed NMR-based titrations of methylated and unmethylated FUS^RGG3PY^ with increasing amounts of EGCG using a fixed concentration of (UG)_12_ RNA. In both titrations, we observed CSPs and line broadening of several ^1^H-^15^N (me)FUS^RGG3PY^ cross-peaks (Figure 3D–F), showing that the presence of RNA is compatible with EGCG binding to the (me)FUS^RGG3PY^. The CSPs of ^1^H-^15^N FUS^RGG3PY^ cross-peaks increase linearly with increasing EGCG concentration for unmethylated FUS (Figure 3G), as observed before in the absence of RNA, suggesting a non-specific binding. On the contrary, we observed specific binding of EGCG to meFUS^RGG3PY^ in the presence of RNA in the low micromolar concentration range, with an associated dissociation constant (K_D_) of 21.7 ± 9.6 µM (Figure 3G,H). This specific binding of EGCG to the meFUS^RGG3PY^ was not observed in the absence of RNA (Figure 2E), suggesting that multiple binding modes involving RNA and methylated arginines are the key molecular determinants to provide specific binding of EGCG to FUS^RGG3PY^.

### 3.4. EGCG Binds to meFUS^RGG3PY^ Involving Its Methylated Arginines

It has been previously found that methylation of FUS^RGG3PY^ by PRMT1 generates asymmetric dimethylarginine (ADMA) and monomethylarginine (MMA) by adding two or one methyl group, respectively, to one of the nitrogen atoms of the arginine guanidinium group [38]. Here, we observed the ADMA/MMA methyl protons as additional peaks in the ^1^H NMR spectra (Figure 4A). To investigate whether EGCG can directly bind to these methyl groups, we performed a titration of meFUS^RGG3PY^ with increasing concentrations of EGCG and recorded sets of ^1^H NMR spectra. Using published EGCG ^1^H resonance assignments [39], we identified chemical shift perturbations of several EGCG protons (Figure 4B). In line with EGCG binding observed in ^1^H-^15^N HSQC spectra (Figure 2C), we observed progressively increased CSPs of several ^1^H resonances with increasing concentrations of EGCG. These CSPs are associated with a consecutive decrease in peak intensities for ^1^H signals corresponding to aromatic protons of phenylalanine (Figure 4C), arginine side-chain protons (Figure 4C) and the methyl groups (Figure 4D). To test whether direct contacts between meFUS^RGG3PY^ and EGCG could be observed, we recorded a 2D ^1^H-^1^H NOESY spectrum. In agreement with the CSPs, we observed a cross-peak between the EGCG aromatic protons and ADMA/MMA methyl protons (Figure 4E). In addition, a ^1^H-^1^H cross-peak corresponding to contact between ADMA/MMA methyl groups and phenylalanine aromatic protons was observed. Altogether, our data indicate that binding of EGCG to meFUS^RGG3PY^ is mediated by its methylated arginine side chains.

### 3.5. meFUS^RGG3PY^ Phase Separation Is Restored by EGCG

It has been previously demonstrated that the RGG3-PY region of FUS mediates phase separation and contributes to FUS stress granule association and the formation of (patho)physiological aggregates in cells [23,27,28,29]. We, therefore, proceeded to assess whether binding of EGCG to FUS^RGG3PY^ can modulate its propensity to phase separate in vitro. Taking into consideration that EGCG interacts with both methylated and unmethylated FUS, we performed turbidity assays as a measure of FUS phase separation for both FUS^RGG3PY^ and meFUS^RGG3PY^ with increasing EGCG concentrations. For both methylated and unmethylated FUS, we observed comparable effects of EGCG: addition of 5 to 100 µM EGCG did not affect FUS^RGG3PY^ turbidity, whereas increasing EGCG concentrations from 100 to 500 µM led to a progressive turbidity increase. This shows that EGCG can lead to the formation of FUS condensates at high EGCG concentrations and that methylation slightly enhances EGCG-dependent FUS condensate formation (Figure 5A).

We previously showed that the presence of RNA can drive phase separation of the FUS^RGG3PY^ and that methylation of full-length FUS impairs FUS LLPS in vitro and stress granules association in cells [23]. Therefore, we proceeded to determine the influence of EGCG on the ability of methylated and unmethylated FUS^RGG3PY^ to phase separate in the presence of RNA. Previous studies identified a higher LLPS propensity of unmethylated FUS^RGG3PY^ with the presence of RNA [23]. We, therefore, first validated that the same observation could be observed in our ratio of protein/RNA concentration. Indeed, whereas stepwise addition of (UG)_12_ RNA led to a progressive increase in turbidity of FUS^RGG3PY^, it did not affect meFUS^RGG3PY^ turbidity (Figure 5B), thus, showing that PRMT1-mediated methylation of the FUS RGG3-PY region completely impairs its RNA-driven propensity to phase separate at our experimental conditions.

Next, we aimed to unravel the effect of EGCG on RNA-driven FUS^RGG3PY^ LLPS. Consequently, we used a fixed concentration of (UG)_12_ RNA, at which FUS^RGG3PY^ showed phase separation, and added increasing concentrations of EGCG to FUS^RGG3PY^ or meFUS^RGG3PY^, respectively. For unmethylated FUS^RGG3PY^, the turbidity remained unchanged up to concentrations of 100 µM EGCG (Figure 5C). This remains in line with our observation that in this specific concentration range, EGCG had no effect on FUS^RGG3PY^ turbidity (Figure 5A). Further addition of EGCG up to 500 µM led to a progressive increase in turbidity, likely due to an RNA-independent effect of EGCG, as also shown in Figure 5A. In contrast, for methylated FUS^RGG3PY^, we observed a progressive increase in FUS^RGG3PY^ turbidity already after the addition of 5 to 100 µM EGCG. Given that neither RNA nor EGCG alone (0–100 µM) had any impact on methylated FUS turbidity, this observation strongly suggests that EGCG restores phase separation of meFUS^RGG3PY^ in the presence of RNA.

In order to validate these striking findings, we performed differential contrast microscopy (DIC) of FUS^RGG3PY^ and meFUS^RGG3PY^ in the presence or absence of 7.5 µM RNA and in the presence or absence of 25 µM EGCG, respectively. In line with the turbidity assays, no condensate formation could be observed for both FUS^RGG3PY^ and meFUS^RGG3PY^ after the addition of 25 µM EGCG (Figure 5D), confirming that at this low EGCG concentration, EGCG has no effect on FUS phase separation. For unmethylated FUS^RGG3PY^, the addition of 7.5 µM (UG)_12_ RNA led to the formation of small condensates that increased in size and number over time, as previously described [23] (Figure 5E, left panel). Further addition of EGCG did not change RNA-driven formation of FUS^RGG3PY^ droplets (Figure 5E, right panel), suggesting that EGCG has no effect on phase separation of the unmethylated FUS RGG3-PY region. In contrast, whereas no condensates could be observed for meFUS^RGG3PY^ in the presence of RNA (Figure 5F, left panel), addition of EGCG led to the formation of small condensates that increased in number over time (Figure 5F, right panel), showing that both the presence of RNA and EGCG are required to allow phase separation of the methylated version of the FUS RGG3-PY region.

To summarize, EGCG binding to unmethylated FUS^RGG3PY^ has no effect on its propensity to phase separate in the presence of RNA. In contrast, while FUS^RGG3PY^ phase separation in the presence of RNA is impaired upon PRMT1-mediated FUS arginine methylation, it is restored by EGCG.

## 4. Discussion

In this study, we found that EGCG directly binds to the RGG3-PY region of FUS, with a non-specific/low-affinity behavior as observed by a linear increase in CSPs of FUS ^1^H-^15^N cross-peaks upon EGCG addition. Despite the non-specific trend of binding, we observed that EGCG seems to preferentially bind to the RG/RGG stretches within the FUS RGG3-PY region, as shown by the overall higher CSPs of these regions compared with non-RG-rich regions (Figure 1C, Figure 2D and Figure 3F). The presence of aromatic rings in EGCG could mediate cation–π interactions with arginines of RG/RGG repeats. Several studies have demonstrated that EGCG can directly interact with the intrinsically disordered regions (IDRs) of various proteins in vitro [21,40,41], including amyloidogenic proteins such as tau, α-synuclein, amyloid-β, but also transcription factors such as p53 [42]. As the corresponding binding regions are highly divergent in terms of amino acid composition, it remains questionable whether EGCG interacts with disordered regions in a non-specific manner at high concentrations. In this regard, we also observed that EGCG can directly interact with the IDRs of axin-1 and FOXO4, two other non-amyloidogenic proteins, at high concentrations (Appendix A
Figure A2).

In contrast, we observed that PRMT1-mediated methylation of the FUS RGG3-PY region in association with the presence of RNA allows specific binding of EGCG with low micromolar affinity. We showed that RNA can also directly interact with meFUS^RGG3PY^, but it remains unclear how RNA binding to methylated RG/RGG regions in FUS could create a specific binding interface for EGCG. Binding of methylated arginines to Tudor domains that act as “methylation readers” has been described recently [43]. In protein complex structures, it was observed that binding of ADMA to Tudor domains is mediated by cation–π interactions involving conserved aromatic cages. We speculate that the three aromatic rings of EGCG resemble the properties of the conserved aromatic cage of Tudor domains containing three aromatic rings and, through this, mediate the interaction with methylated arginines. Further structural studies will be required to address this and how the number and patterns of methylation regulate EGCG binding. Nevertheless, we showed that the methyl groups of methylated arginines directly contact EGCG (Figure 4D,E). RG/RGG regions are highly abundant in the human proteomes. Many of these regions have been shown to be methylated as well as to bind RNA [36,44]. Therefore, it is tempting to speculate that EGCG binding observed for FUS RGG3-PY here could be relevant for other proteins containing RG/RGG regions.

EGCG function in neuroprotection is primarily associated with its inhibitory effect on amyloidogenic protein aggregation, redirecting the self-association of amyloidogenic peptides from toxic aggregates/amyloid fibrils to stable, non-toxic oligomers [18,21,45,46]. Numerous RG/RGG-containing proteins have been implicated in neurological disorders, such as FUS and TAF-15 in ALS, FXR1, and FMRP1 in fragile X syndrome (FSX), SMN in spinal muscular atrophy (SMA), SHANK1 in autism, as well as a plethora of human cancers [44]. Here, we showed that specific EGCG binding to the RG/RGG region of FUS allows condensates formation of methylated FUS in the presence of RNA. Methylation of FUS RG/RGG regions as well as several other disease-relevant RBPs attenuates phase separation in vitro and SG association in cells and is currently seen as a protective role against the redirection of physiological “liquid-like” membrane-less organelles to toxic aggregates and fibrils [23,36]. Therefore, it remains to be determined whether the observed effect of EGCG on phase separation of RG/RGG regions has beneficial or adverse effects on RG/RGG-linked diseases.

Our data propose unexpected new functions of EGCG in the regulation of biomolecular condensate formation, targeting one of the most abundant regions in the human proteome. Through this, our work might set the base to promote a better understanding of the EGCG effect in neuroprotection, as well as putative other disorders linked to misregulation of RG/RGG-containing proteins.

## Figures and Tables

**Figure 1 cells-11-00592-f001:**
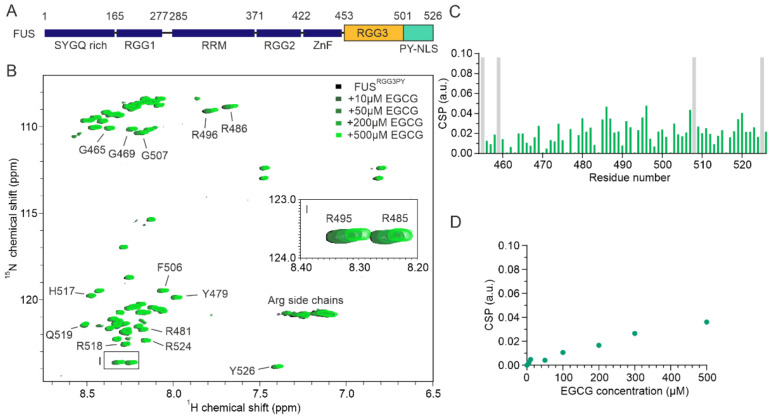
FUS^RGG3PY^ binds directly to EGCG. (**A**) Scheme of the FUS domain organization; SYGQ-rich, RRM (RNA-recognition motif), ZnF (Zinc finger), PY-NLS (proline-tyrosine nuclear localization signal), arginine-glycine/glycine (RG/RGG). (**B**) Overlay of ^1^H-^15^N HSQC spectra of 30 µM ^15^N-labeled FUS^RGG3PY^ in the absence (black) and presence of the increasing concentrations of EGCG, ranging from 10 to 500 µM (colored in a gradient of green). The region corresponding to arginine R485 and R495 ^1^H-^15^N HSQC cross-peaks is represented as a zoomed-in image on the middle right side of the spectrum. (**C**) Chemical shift perturbation (CSP) plot of FUS^RGG3PY 1^H-^15^N HSQC cross-peaks in the presence of 500 µM EGCG (samples from (**B**)). Proline residues lacking a backbone amide signal are indicated in gray. (**D**) CSP plot of a FUS^RGG3PY 1^H-^15^N cross-peak corresponding to R495 plotted as a function of EGCG concentration (samples from (**B**)).

**Figure 2 cells-11-00592-f002:**
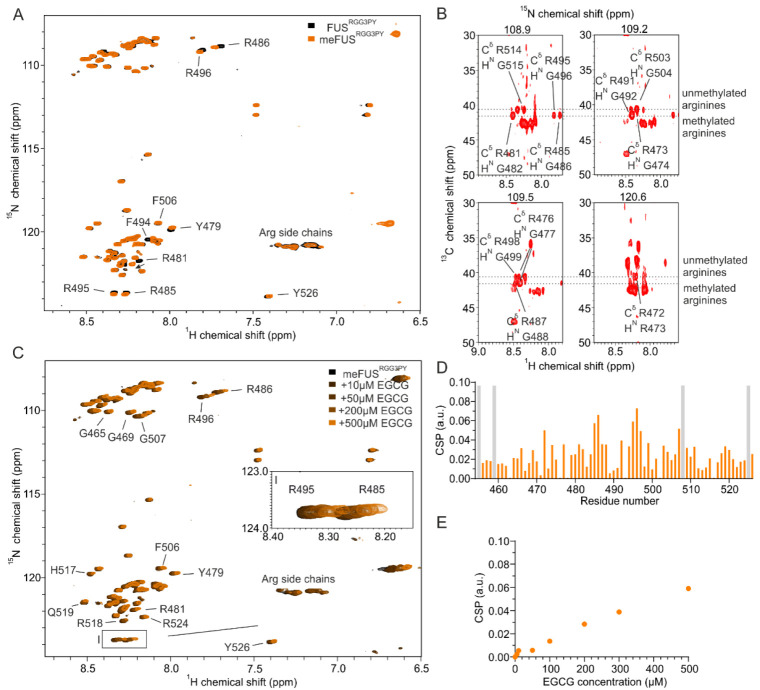
Arginine methylation of FUS^RGG3PY^ and EGCG binding of meFUS^RGG3PY^. (**A**) Overlay of ^1^H-^15^N HSQC spectra of 30 µM ^15^N-labeled FUS^RGG3PY^ (black) and PRMT1-methylated meFUS^RGG3PY^ (orange). (**B**) Strip plots of (H)CC(CO)NH NMR spectra corresponding to the Cδ atoms of arginines of PRMT1-methylated FUS^RGG3PY^. ^13^Cδ of ^1^H-^13^C cross-peaks corresponding to methylated and non-methylated arginine residues are indicated by dotted lines. (**C**) Overlay of ^1^H-^15^N HSQC spectra of 30 µM ^15^N-labeled meFUS^RGG3PY^ in the absence (black) and presence of increasing concentrations of EGCG, ranging from 10 to 500 µM (colored in a gradient of orange). The region corresponding to R485 and R495 ^1^H-^15^N HSQC cross-peaks is shown as a zoomed-in image on the middle right side of the spectrum. (**D**) CSP plot of meFUS^RGG3PY 1^H-^15^N HSQC cross-peaks in the presence of 500 µM EGCG (samples from (**C**)). Proline residues lacking a backbone amide signal are shown in gray. (**E**) CSP plot of a meFUS^RGG3PY 1^H-^15^N cross-peak corresponding to R495 plotted as a function of EGCG concentration (samples from (**C**)).

**Figure 3 cells-11-00592-f003:**
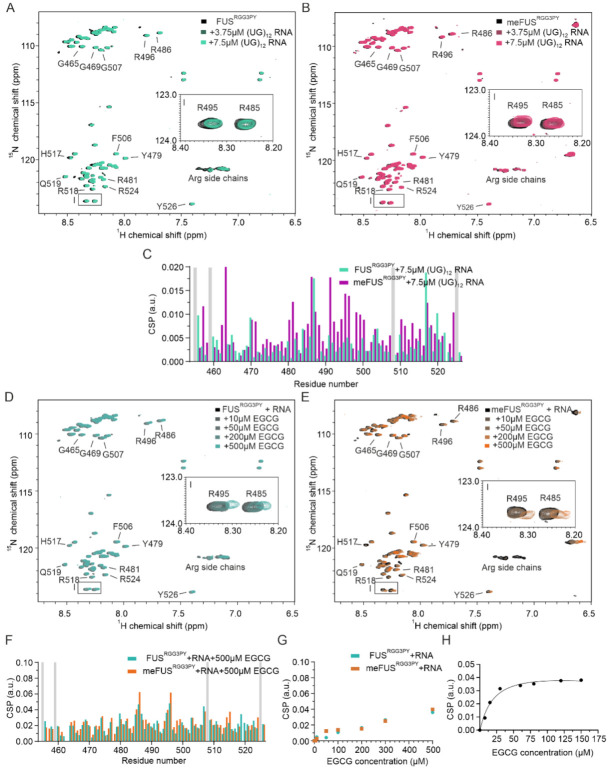
Role of RNA in FUS^RGG3PY^ and meFUS^RGG3PY^ -EGCG binding. (**A**,**B**) Overlays of ^1^H-^15^N HSQC spectra of 30 µM ^15^N-labeled FUS^RGG3PY^ (**A**) and 30 µM ^15^N-labeled meFUS^RGG3PY^ (**B**) in the absence (black) and presence of the increasing concentrations of (UG)_12_ RNA, at 0, 3.75, and 7.5 µM (colored in a gradient of green for the samples of FUS^RGG3PY^, and in a gradient of magenta for the samples of meFUS^RGG3PY^). (**C**) Overlay of CSP plots of FUS^RGG3PY^ (green) and meFUS^RGG3PY^ (magenta) ^1^H-^15^N HSQC cross-peaks in the presence of 7.5 µM (UG)_12_ RNA (samples from (**A**), (**B**)). Proline residues lacking a backbone amide signal are indicated in gray. (**D**,**E**) Overlay of ^1^H-^15^N HSQC spectra of 30 µM ^15^N-labeled FUS^RGG3PY^ (**D**) and 30 µM ^15^N-labeled meFUS^RGG3PY^ (**E**) in the presence of 7.5 µM (UG)_12_ RNA. The ^1^H-^15^N HSQC spectra recorded at increasing concentrations of EGCG, ranging from 10 to 500 µM, are colored in a gradient of blue and orange for the samples of FUS^RGG3PY^ and meFUS^RGG3PY^, respectively. The regions corresponding to R485 and R495 ^1^H-^15^N HSQC cross-peaks are indicated by zoomed-in images on the middle right side of the spectra. (**F**) Overlay of CSP plots of ^1^H-^15^N HSQC cross-peaks of FUS^RGG3PY^ (blue) and meFUS^RGG3PY^ (orange), in the presence of 7.5 µM (UG)_12_ RNA and 500 µM EGCG (samples from (**D**,**E**)). Proline residues lacking a backbone amide signal are shown in gray. (**G**) CSP plots of FUS^RGG3PY^ (blue) and meFUS^RGG3PY^ (orange), both in the presence of 7.5 µM (UG)_12_ RNA. ^1^H-^15^N cross-peaks corresponding to R495 are plotted as a function of EGCG concentration (samples from (**D**,**E**)). (**H**) CSP plot of a ^1^H-^15^N cross-peak corresponding to R487 of meFUS^RGG3PY^ in the presence of 7.5 µM (UG)_12_ RNA plotted for increasing concentrations of EGCG. The associated K_d_ is estimated to be 21.7 ± 9.6 µM.

**Figure 4 cells-11-00592-f004:**
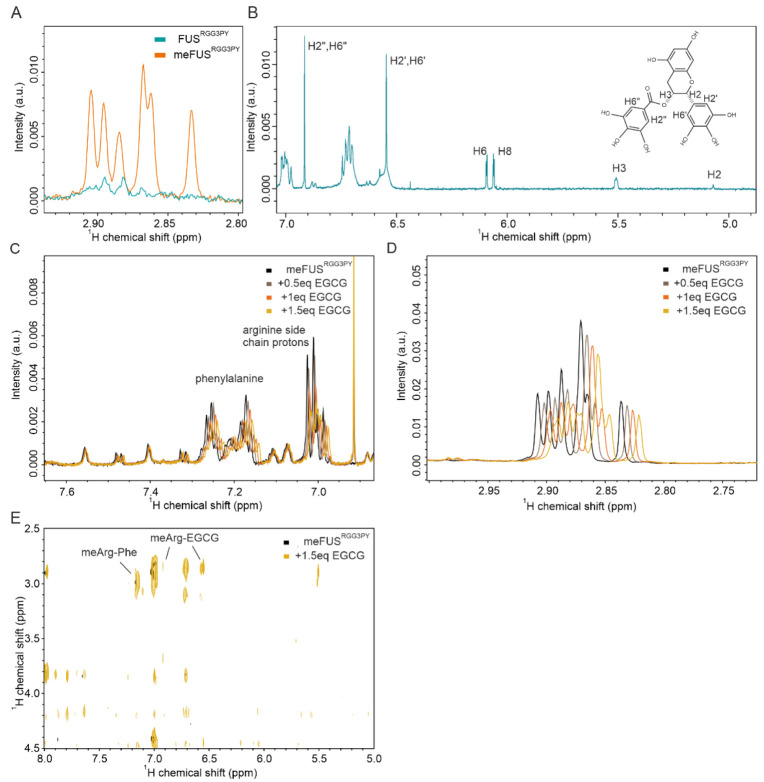
EGCG binds to the side chains of methylated arginines in meFUS^RGG3PY^. (**A**) Overlay of ^1^H spectra of FUS^RGG3PY^ (light blue) and meFUS^RGG3PY^ (orange) showing the chemical shift region corresponding to ADMA and MMA. (**B**) ^1^H NMR spectrum of 100 µM ^15^N-labeled meFUS^RGG3PY^ in the presence of 1.5 stoichiometric equivalents of EGCG. The structural formula and the EGCG protons that were assigned are marked on the right. The proton numbering refers to the published assignment [39]. (**C**) CSPs and a decrease in intensity of signals corresponding to phenylalanine and arginine side-chain protons were observed with increasing concentrations of EGCG at 0.5 (gray), 1.0 (orange), and 1.5 (yellow) stoichiometric equivalents. (**D**) CSPs and a decrease in intensity of signals corresponding to ADMA and MMA were observed with increasing concentrations of EGCG at 0.5 (gray), 1.0 (orange), and 1.5 (yellow) stoichiometric equivalents. (**E**) Overlay of ^1^H-^1^H NOESY spectra of meFUS^RGG3PY^ (black) and meFUS^RGG3PY^ in the presence of 1.5 stoichiometric equivalents of EGCG (yellow) showing cross-peaks between ADMA/MMA methyl/phenylalanine aromatic protons and EGCG.

**Figure 5 cells-11-00592-f005:**
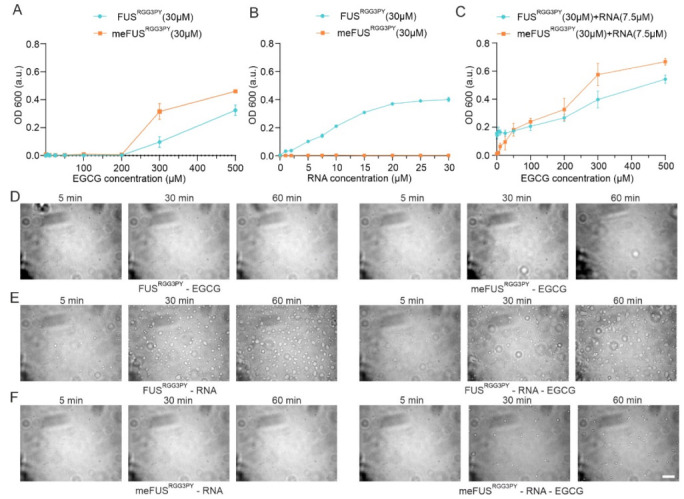
Turbidity and DIC microscopy studies of FUS^RGG3PY^ and meFUS^RGG3PY^ liquid-liquid phase separation. (**A**,**B**) Turbidity assays were performed at a fixed concentration of FUS^RGG3PY^ and meFUS^RGG3PY^ (both at 30 µM) with an increasing concentration of EGCG (**A**) or (UG)_12_ RNA (**B**). (**C**) Turbidity assays for FUS^RGG3PY^ and meFUS^RGG3PY^ (at a fixed concentration of 30 µM), both in the presence of 7.5 µM (UG)_12_ RNA, with increasing concentrations of EGCG. Data points for the methylated FUS^RGG3PY^ are shown in orange and for the non-modified protein in blue. (**D**) Differential interference contrast microscopy images illustrating FUS^RGG3PY^ and meFUS^RGG3PY^ at 30 µM in the presence of 25 µM EGCG are shown on the left and the right panels, respectively. (**E**) DIC images of 30 µM FUS^RGG3PY^ in the presence of 7.5 µM (UG)_12_ RNA and in the absence (left panel) or presence of 25 µM EGCG (right panel). (**F**) DIC images of 30 µM meFUS^RGG3PY^ in the presence of 7.5 µM (UG)_12_ RNA in the absence (left panel) or presence of 25 µM EGCG (right panel). All images were recorded over 1 h, and the scale bar represents 10 µm.

## Data Availability

The data shown in this study are available upon request from the corresponding author. The recorded NMR data are available at Biological Magnetic Resonance Bank under the accession codes 51314 and 51315 for FUS^RGG3PY^ and metFUS^RGG3PY^, respectively.

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
