# Peer review of "EGCG Promotes FUS Condensate Formation in a Methylation-Dependent Manner"

_cells, 2022, doi:10.3390/cells11040592_

Round 1

Reviewer 1 Report

The work presented and particularly rigorous. The subject deals with an important medical issue, neurodegenerative diseases. Using derivatives known to have a neuroprotective role, the authors perform work of interest. Tthe only general downside of the article is the confusion between the applications of its results in the clinic, the fact is that this approach is currently far from a short-term application. It would be good to highlight the limitations of the approach compared to the proposal of a drug or any dosage. Indeed, the neophyte reader can be deceived by the approach.

The introduction and very pleasant to read it would just be interesting to explain the limitations of the compounds presented. The second weight would be to specify the critical interest of FUS as a model protein.

The material and method is particularly rich and well documented. It corresponds perfectly to what one expects from this type of section.

Only one fact remains particularly unclear for me, it is the mix between arginines not methylated, partially and completely methylated. Do the results depend on this second category or is it just present but not acting.

Another point, in the figures of the gray bars shows residue a one has signed comma this in white correspond to another state?

The protein is quite long and the work is focused on a specific area. However, the others tomorrow would not have an action or an implication in the observed result?

in figure 5 some panel looks quite the same, is this a reader's view or reality? Shouldn’t we have a little difference (background noise)?

Author Response

The work presented and particularly rigorous. The subject deals with an important medical issue, neurodegenerative diseases. Using derivatives known to have a neuroprotective role, the authors perform work of interest. The only general downside of the article is the confusion between the applications of its results in the clinic, the fact is that this approach is currently far from a short-term application. It would be good to highlight the limitations of the approach compared to the proposal of a drug or any dosage. Indeed, the neophyte reader can be deceived by the approach.

The introduction and very pleasant to read it would just be interesting to explain the limitations of the compounds presented.

We thank the reviewer for her/his positive feedback. To clarify that our study is a basic research project and that we don’t present a clinical study we added a brief statement in the last paragraph of introduction, stating that future studies will be required to fully unravel the physiological role of EGCG-regulated condensate formation and potential applications of EGCG-derived compounds in light of treatment.

The second weight would be to specify the critical interest of FUS as a model protein.

In our manuscript we introduce the biological significance of FUS, as well as proteins harboring RG/RGG regions, in lines 52 to 68. With this paragraph we aim to cover the main aspects of FUS relevant to this study.

The material and method is particularly rich and well documented. It corresponds perfectly to what one expects from this type of section.

The authors thank the reviewer for the appreciation.

Only one fact remains particularly unclear for me, it is the mix between arginines not methylated, partially and completely methylated. Do the results depend on this second category or is it just present but not acting.

In Fig. 2 we show resonance peaks corresponding to arginine residues within the RG/RGG region of meFUSRGG3-PY in three different methylation status: non-methylated, partially methylated (i.e. in some of FUSRGG3PY molecules it’s methylated, but in others not) and completely methylated. The aim was to indicate that not all arginines are fully methylated, and this residue-specific methylation pattern caused the observed EGCG binding and subsequent difference in the propensity of RNA-driven phase separation. However, it remains to be clarified in future studies how patterns and number of methylated arginines impacts EGCG binding and condensate formation. We have added a corresponding statement to the revised version of the manuscript.

Another point, in the figures of the gray bars shows residue a one has signed comma this in white correspond to another state?

In the CSP plot, residues covered with gray bars correspond to proline residues which lack a backbone amide proton and can therefore not be included in the CSP analysis. Residues without gray bar correspond to backbone amides of assigned residues. We updated the figure legends accordingly.

The protein is quite long and the work is focused on a specific area. However, the others tomorrow would not have an action or an implication in the observed result?

We agree with the reviewer that future studies should focus on the role of EGCG binding in context of the full-length protein. In this study, the authors focus on the third RG/RGG region of FUS and used it as a model protein, because RG/RGG regions are widely present in the human proteome, including proteins like CIRBP, TAF15, hnRNPA1 and RBMX as example. These regions are found to be responsible for the formation of condensates via the process of liquid-liquid phase separation arising from multivalent interactions with RNA. Therefore, this indicates that EGCG can bind to RG/RGG regions of other proteins through a similar mechanism, and that this might restore condensates. In the future we aim to address this and the mechanistic implications in cells.

in figure 5 some panel looks quite the same, is this a reader's view or reality? Shouldn’t we have a little difference (background noise)?

Figure 5 shows the change in droplet size and number over time for different FUS versions and in absence/presence of EGCG and/or RNA. Time courses were analyzed in the same samples over several time points (5, 30, 60 minutes). The background is caused by the DIC microscope setup.

Reviewer 2 Report

The authors present new evidence that EGCG binds directly to FUS and meFUS phase separation can be restored in the presence of RNA molecules at low concentration of EGCG. This paper is an important step forward in the understanding of the mechanism by which EGCG binds FUS, however there are few key questions that need to be addressed to fully understand the mechanism.

Specifically, given the presence of multiple Arg residues in EGCG, the authors need parallel approaches to confirm the binding is occurring through one (or more) of them. For example, a compelling experiment is a mutagenesis approach were individual Arg are mutated to Ala to identify the actual aminoacids responsible for binding. A very interesting and strong evidence, would also be testing the presence of Lys residues in place of Arg residues. Lys residue, indeed, can still make a cation-pi interaction with the molecule, but they cannot be methylated by PRMT1. Therefore, this approach will define more clearly which one of the methylated arginines are improving the binding. If the hypothesis that EGCG binds the positive amino acids is true, methylation should abolish this. However, the authors see the opposite result. This cannot be explained by the experiment proposed and it might help looking at the structure of FUS (via AlphaFold2 if the structure is unknown) and see where these residues are located. It might be that the quaternary arrangement of the protein favours the methylated vs. unmethylated version of FUS in binding EGCG. With the current data we don't know what is going on and we need further experiments to address the mechanism.

Minor points:

  • The authors state that there is no specific binding of EGCG to FUS. However, Fig.5 shows a dose-response curve. Even though some points are missing, it seems to me that the effect is dose-dependent, hence specific. The authors should test more concentration points, including higher concentration to saturate the curve. With more points it would be easier to calculate a fitting curve with an actual EC50 value.
  • What the do the authors mean by "in a methylation independent manner "(line 276)? From Fig.5A seems clear that methylation increases affinity.  
  • The authors should clarify how the protein is refolded. From the methods section it is unclear. It is not mentioned specifically but it let the reader assume that the refolding has been done on Ni-NTA column prior to elution. Please advise.

Author Response

The authors present new evidence that EGCG binds directly to FUS and meFUS phase separation can be restored in the presence of RNA molecules at low concentration of EGCG. This paper is an important step forward in the understanding of the mechanism by which EGCG binds FUS, however there are few key questions that need to be addressed to fully understand the mechanism.

We thank the reviewer for her/his positive feedback.

Specifically, given the presence of multiple Arg residues in EGCG, the authors need parallel approaches to confirm the binding is occurring through one (or more) of them. For example, a compelling experiment is a mutagenesis approach were individual Arg are mutated to Ala to identify the actual aminoacids responsible for binding. A very interesting and strong evidence, would also be testing the presence of Lys residues in place of Arg residues. Lys residue, indeed, can still make a cation-pi interaction with the molecule, but they cannot be methylated by PRMT1. Therefore, this approach will define more clearly which one of the methylated arginines are improving the binding. If the hypothesis that EGCG binds the positive amino acids is true, methylation should abolish this. However, the authors see the opposite result. This cannot be explained by the experiment proposed and it might help looking at the structure of FUS (via AlphaFold2 if the structure is unknown) and see where these residues are located. It might be that the quaternary arrangement of the protein favours the methylated vs. unmethylated version of FUS in binding EGCG. With the current data we don't know what is going on and we need further experiments to address the mechanism.

Indeed, our hypothesis is that arginine methylation enhances EGCG binding to the RG/RGG region of FUS. At the moment, we don’t know if specific patterns of arginine methylation determine EGCG binding or whether the overall number of arginine methylation is the key determinant for binding.

Arginine methylation retains the positive charge, and it is interesting to mention that folded proteins binding methylated arginines share a characteristic hydrophobic aromatic core which might share similar properties with aromatic rings in EGCG (see discussion). It is indeed our future aim to identify the molecular details of EGCG binding and the exact role of methylation patterns and the degree of methylation.

FUSRGG3PY is intrinsically disordered and does not adopt a stable tertiary structure. To further support this, we have included additional heteronuclear NOE NMR data (appendix figure A1). FUS residues show low or negative heteronuclear NOEs indicating that the protein is disordered in both, its unmethylated and methylated state.

Minor points:

The authors state that there is no specific binding of EGCG to FUS. However, Fig.5 shows a dose-response curve. Even though some points are missing, it seems to me that the effect is dose-dependent, hence specific. The authors should test more concentration points, including higher concentration to saturate the curve. With more points it would be easier to calculate a fitting curve with an actual EC50 value.

We interpreted the increase of turbidity observed in Figure 5 also in light of the NMR data which showed that i) binding could not be saturated even at the highest concentrations of EGCG used (XXX mM) and that ii) EGCG binds to completely unrelated disordered proteins Axin-1 and FOXO4 (Figure A2). Specific EGCG binding was observed consistently for meFUSRGG3PY/RNA, as supported by saturation of the NMR titration (Figure 3H) and the steep increase of turbidity and plateauing of turbidity at around 100 µM EGCG (Figure 5C). Nevertheless, at higher concentrations (above 300 µM), a second, probably unspecific binding mode seems to be involved.

What do the authors mean by "in a methylation independent manner "(line 276)? From Fig.5A seems clear that methylation increases affinity. 

We apologize for the confusion raised. Indeed, the higher turbidity observed for methylated FUS upon EGCG addition might indicate enhanced formation of FUS condensates. We updated the text accordingly.

The authors should clarify how the protein is refolded. From the methods section it is unclear. It is not mentioned specifically but it let the reader assume that the refolding has been done on Ni-NTA column prior to elution. Please advise.

FUSRGG3PY is an intrinsically disordered region and does not adopt a stable tertiary fold. The purification was carried out using urea in order to prevent proteolytic degradation of the protein. Urea was removed prior TEV cut and is absent in any buffers used for the experiments shown in the manuscript.

Round 2

Reviewer 2 Report

The authors answered to all my concerns